# Association between statin use and clinical course, microbiologic characteristics, and long-term outcome of early Lyme borreliosis. A post hoc analysis of prospective clinical trials of adult patients with erythema migrans

**Daša Stupica**[1,2]*, **Fajko F. Bajrović**[3,4], **Rok Blagus**[5,6], **Tjaša Cerar Kišek**[7], **Stefan Collinet-Adler**[8], **Eva Ružić-Sabljić**[7], **Maša Velušček**[1]

1 Department of Infectious Diseases, University Medical Center Ljubljana, Ljubljana, Slovenia, 2 Faculty of Medicine, University of Ljubljana, Ljubljana, Slovenia, 3 Institute of Pathophysiology, Faculty of Medicine, University of Ljubljana, Ljubljana, Slovenia, 4 Department of Neurology, University Medical Center Ljubljana, Ljubljana, Slovenia, 5 Institute for Biostatistics and Medical Informatics, Faculty of Medicine, University of Ljubljana, Ljubljana, Slovenia, 6 Faculty of Sports, University of Ljubljana, Ljubljana, Slovenia, 7 Institute for Microbiology and Immunology Ljubljana, Faculty of Medicine, University of Ljubljana, Ljubljana, Slovenia, 8 Department of Infectious Diseases, Park Nicollet/Health Partners, Methodist Hospital, Saint Louis Park, Minnesota, United States of America

* dasa.stupica@kclj.si

## Abstract

### Background

Statins were shown to inhibit borrelial growth *in vitro* and promote clearance of spirochetes in a murine model of Lyme borreliosis (LB). We investigated the impact of statin use in patients with early LB.

### Methods

In this post-hoc analysis, the association between statin use and clinical and microbiologic characteristics was investigated in 1520 adult patients with early LB manifesting as erythema migrans (EM), enrolled prospectively in several clinical trials between June 2006 and October 2019 at a single-center university hospital. Patients were assessed at enrollment and followed for 12 months.

### Results

Statin users were older than patients not using statins, but statin use was not associated with *Borrelia* seropositivity rate, *Borrelia* skin culture positivity rate, or disease severity as assessed by erythema size or the presence of LB-associated symptoms. The time to resolution of EM was comparable in both groups. The odds for incomplete recovery decreased with time from enrollment, were higher in women, in patients with multiple EM, and in those reporting LB-associated symptoms at enrollment, but were unaffected by statin use.

**Data Availability Statement:** The minimal dataset required to replicate the reported study findings are within the paper. Additional data cannot be shared publicly because of potentially sensitive patient information. A non-author contact that interested researchers can get in touch with in terms of accessing the data is Prof. Tatjana Lejko Zupanc, Head of the Department of Infectious Diseases, University Medical Center Ljubljana, who can be reached at tatjana.lejko@kclj.si.

**Funding:** This research was supported by the Slovenian Research Agency [grant number P3-0296, J3-6788]. The funders had no role in study design, data collection and analysis, decision to publish, or preparation of the manuscript.

**Competing interests:** The authors have no conflicts of interest to declare that are relevant to the content of this article. The funders had no role in the design of the study; in the collection, analyses, or interpretation of data; in the writing of the manuscript, or in the decision to publish the results. Some of the information was presented at the International Symposium on Tick-Borne Pathogens and Disease ITPD 2019, September 2019, Vienna, Austria. This does not alter our adherence to PLOS ONE policies on sharing data and materials

## Conclusion

Statin use was not associated with clinical and microbiologic characteristics or long-term outcome in early LB.

## Introduction

Erythema migrans (EM) is the most common manifestation of early Lyme borreliosis (LB), caused by the tick-borne spirochete *Borrelia burgdorferi* sensu lato (s.l.) [1]. Manifestations of LB usually resolve without sequelae when well-studied, safe and effective antibiotic regimens are used [1, 2]. Still, in a small minority of cases, post-LB symptoms such as malaise, fatigue, neurocognitive dysfunction and diffuse pain, may persist for months to years despite otherwise microbiologically successful antibiotic treatment [1–3]. Adjunctive treatment targeting immune/inflammatory and metabolic responses to *Borrelia* infection could conceivably improve outcomes in this subset of patients.

Statins are widely used to treat hypercholesterolemia and reduce the risk of cardiovascular and cerebrovascular diseases [4]. They have a generally acceptable safety profile and can be cost-effective [4, 5]. Statins inhibit 3-hydroxy-3-methylglutaryl-coenzyme A reductase (HMGR), a major regulatory and rate limiting enzyme involved in the mevalonic pathway [8]. In eukaryotes, the mevalonic pathway is involved in biosynthesis of cholesterol and other iso-prenoids that are important for cellular composition, interactions, and messaging [6]. In addition to lowering lipids, statins have pleiotropic effects reflecting anti-inflammatory, immunomodulatory, and bactericidal properties [7]. *B. burgdorferi* s.l. has a functional homo-log of the eukaryotic HMGR-I (HMGR-II) [8–10], but biosynthesis of cholesterol through the borrelial mevalonic pathway does not occur and therefore host-derived cholesterol is required for growth [11]. Simvastatin and lovastatin were shown to inhibit borrelial growth under *in vitro* conditions by inhibiting HMGR [8]. In a murine model of LB, statins were shown to reduce bacterial load and alter the immune response to favor clearance of borreliae [12]. It is not known whether the anti-borrelial effects of statins found in *in vitro* conditions and in animal models are applicable to humans, or if hyperlipidemia is associated with progression of borrelial infection in humans. Statin pretreatment was recently found to not be associated with clinical manifestations, laboratory test results, and outcomes in patients with Bannwarth's syndrome [13].

Our study investigated whether statin use is associated with clinical and microbiologic characteristics or long-term outcome in patients treated with antibiotics for early LB manifesting as EM.

## Materials and methods

### Setting and patients

The 1520 patients included in this post-hoc analysis were ≥18 years old with EM, and were enrolled prospectively in several other clinical trials between June 2006 and October 2019 at the University Medical Center in Ljubljana, Slovenia [3, 14–16]. EM was defined according to European criteria [17] as an expanding erythema with or without central clearing, developing days to weeks after a tick bite or exposure to ticks in a LB endemic region. For a reliable diagnosis, the erythema must have reached ≥5 cm in diameter. If the diameter was smaller, a history of tick bite, a delay in appearance of at least two days, and an expanding erythema at the

bite site were required [17]. Multiple EM was defined as the presence of two or more skin lesions, at least one of which had to fulfill the size criterion for solitary EM [18]. At enrollment, patients were prescribed antibiotics according to treatment guidelines [2]. We defined statin use as consistent statin therapy for hyperlipidemia and/or comorbidities requiring primary or secondary prophylaxis of cardiovascular disease at the time of enrollment, followed by ongoing use for the duration of the study.

## Evaluation of patients

History, medication reconciliation and physical examination were performed at baseline and at follow-up (14 days and 2, 6, and 12 months). In addition, patients were asked open-ended questions about health-related symptoms that had newly developed or worsened since the onset of the EM. If these symptoms had no other medical explanation, they were considered LB-associated constitutional symptoms at enrollment or post-LB symptoms at follow-up.

Complete recovery was defined as a return to pre-LB health status. Incomplete recovery was defined as the presence of post-LB symptoms and/or the appearance of new objective signs of LB and/or persistence of borreliae as detected by culture of a re-biopsied skin sample, and/or persistence of EM ≥ 2 months after treatment. Persistence of EM was defined as EM still visible in daylight and at room temperature.

## Laboratory analyses

Serologic data were obtained either by indirect chemiluminescence immunoassay (IgM antibodies to OspC and VlsE, IgG antibodies to VlsE borrelial antigens; LIAISON, Diasorin, Italy), the C6 Lyme ELISA kit (IgM and IgG antibodies to C6 peptide derived from VlsE; Immunetics®, Oxford Immunotec, Marlborough, MA, USA), or by an immunofluorescence assay with a local skin isolate of *B. afzelii* as antigen [19]. Results were interpreted according to the manufacturers' instructions or as titers for the immunofluorescence assay, with titers ≥ 1:128 considered positive.

At the baseline visit, a skin biopsy was taken at the leading edge of the primary EM and placed in 6 mL of a modified Kelly-Pettenkofer culture medium (MKP). If the first skin specimen was culture positive for borreliae, a second skin biopsy was taken from the same site 2–3 months after the start of antibiotic therapy. Baseline blood samples were cultured for borreliae as previously described [20]. Isolates were identified to the species level using pulsed-field gel electrophoresis after MluI restriction of genomic DNA, PCR-based restriction fragment length polymorphism of the intergenic region [21], or real-time PCR targeting the hbb (U48676.1) gene [22].

## Statistical methods

Categorical data were summarized as frequencies (%) and numerical data as medians (interquartile range, IQR). Differences between the groups taking versus not taking statins were tested by the Mann-Whitney test or the chi-square test with Yates continuity correction. The association between the groups taking versus not taking statins and the proportion of patients with incomplete recovery at each follow-up time point was tested using the chi-square test with Yates continuity correction. The median duration of the EM was calculated employing the Kaplan–Meier method; the log-rank test was applied to test the difference between the duration curves in the groups taking versus not taking statins. The association between incomplete recovery and a prespecified set of covariates (patients' sex and age, presence of comorbidities, statin use, presence of LB-associated constitutional symptoms at enrollment, presence of multiple EM, and time from enrollment) was estimated using multiple logistic regression. To

account for multiple measurements in each patient and participants from five different studies, the analysis was also adjusted for a subject variable and a study variable as random effects. Results are presented as odds ratios (OR) with 95% confidence intervals (CI). R statistical language (version 3.4.1) was used for the analyses [23].

### Ethics

The study was carried out in concordance with the Declaration of Helsinki and was approved by the Medical Ethics Committee of the Ministry of Health of the Republic of Slovenia (No. 0120-161/2017/16 and 0120-670/2017/10). Many patients in the present study were also enrolled in other studies conducted at the University Medical Center in Ljubljana. Therefore, this and our previous studies share basic methodologic approaches to clinical and microbiologic evaluation and follow-up [3, 14–16, 24]. All of these studies were approved by the Medical Ethics Committee of the Ministry of Health of the Republic of Slovenia (No. 38/05/06, 83/05/10, 36/05/09, 127/06/10, and 0120-161/2017-5), four of the studies were registered at http://clinicaltrials.gov (identifier NCT03584919, NCT01163994, NCT00910715, and NCT03956212). Written informed consent was obtained from all subjects involved in each of these studies.

## Results

### Patients' characteristics at enrollment

Among the 1520 enrolled patients, 122 (8.0%) were receiving statins at enrollment and continued to do so during follow-up, while 1398 (92.0%) patients were not (Table 1). Patients' pre-treatment characteristics according to statin use are shown in Table 1. The majority (1325, 87.2%) of patients presented with solitary EM while 195 (12.8%) patients had disseminated disease manifesting as multiple EM. There was no significant difference in the frequency of multiple EM in patients taking statins compared to those not on statins (10/122, 8.3% vs. 185/1398, 13.2%; $p = 0.146$).

Statin-users were older (Table 1) and more likely to report having comorbidities other than hyperlipidemia, such as hypertension, osteoporosis, diabetes mellitus, thyroid disease, cardiac arrhythmias, psychiatric disorders, ischemic heart disease, osteoarthritis, or asthma (107/122, 87.7% vs. 523/1398, 37.4%; $p < 0.001$). Overall, patients with solitary EM reported LB-associated constitutional symptoms at enrollment less often than patients with multiple EM (361/1325, 27.2% vs. 92/195, 47.2%; $p < 0.001$). The association between statin use and reported LB-associated constitutional symptoms was not significant. Also, local progression of borrelial infection as assessed by skin erythema size at enrollment was comparable regardless of statin use (Table 1).

### Microbiologic results according to statin use

The seropositivity rate at enrollment was comparable between patients with positive skin and/or blood cultures and those with negative cultures: 376/556, 67.6% vs. 327/498, 65.7% when using the chemiluminescence immunoassay ($p = 0.542$); 10/93, 10.8% vs. 18/111, 16.2% when using the immunofluorescence assay ($p = 0.355$); and 47/63, 74.6% vs. 35/56, 62.5% with the C6 Lyme ELISA ($p = 0.220$). Among the identified *Borrelia* species skin isolates, *B. afzelii* was the most common (91.8%). There was no association between statin use and serologic response to infection or borreliae isolation from skin or blood.

**Table 1. Clinical and microbiologic characteristics of patients with erythema migrans at enrollment according to statin use.**

| Characteristic | Using statin | Not using statin | p Value[a] |
|---|---|---|---|
| | n = 122 | n = 1398 | |
| Male sex | 53 (43.4) | 618 (44.2) | 0.946 |
| Age | 62 (58–69) | 53 (42–61.8) | <0.001 |
| History of Lyme borreliosis | 24 (19.7) | 177 (12.7) | 0.040 |
| Comorbidities[b] | 107 (87.7) | 523 (37.4) | <0.001 |
| Tick bite[c] | 57 (46.7) | 688 (49.2) | 0.665 |
| Days since EM first observed | 11.5 (4–27.3) | 11 (5–27) | 0.829 |
| Diameter of primary EM, cm | 14.5 (10–22) | 15 (10–22) | 0.980 |
| EM with central clearing[d] | 53 (43.4) | 721 (51.6) | 0.103 |
| Multiple EM | 10 (8.2) | 185 (13.2) | 0.146 |
| LB-associated symptoms[e] | 36 (29.5) | 417 (29.8) | 1.000 |
| Seropositive[f] | 73/121 (60.3) | 814/1371 (59.4) | 0.913 |
| Skin culture positive | 56/112 (50.0) | 650/1269 (51.2) | 0.881 |
| B. afzelii | 45/49 (91.8) | 523/586 (89.2) | 0.799[g] |
| B. garinii | 3 (6.1) | 37 (6.3) | |
| B. burgdorferi sensu stricto | 1 (2.0) | 10 (1.7) | |
| Unidentified | 0 (0) | 16 (2.7) | |
| Blood culture positive | 1 (0.8) | 15 (1.1) | 1.000 |

Abbreviations: EM, erythema migrans.

[a]In order to correct for multiple comparisons, $p < 0.01$ was considered significant.

[b]Patients with an underlying chronic illness in addition to hyperlipidemia.

[c]Patients with a history of tick bite at EM site.

[d]In patients with multiple EM, the primary lesion was assessed for central clearing.

[e]Patients who reported constitutional symptoms that had newly developed or worsened since the onset of the EM. Some patients had more than one constitutional symptom.

[f]Positive test result for IgM and/or IgG antibodies to B. burgdorferi sensu lato at enrollment.

[g]Comparison between B. afzelii and other identified Borrelia species. Identification was performed in 49 and 586 Borrelia isolates in the statin and non-statin group, respectively.

## Treatment outcome according to statin use

Twenty-seven of 1520 (1.8%) patients missed the 14-day follow up, with increasing losses to follow-up at subsequent visits (Table 2). The median time to resolution of EM after starting

**Table 2. Number (%) of patients with erythema migrans who had incomplete recovery at follow-up visits according to using or not using statins.**

| | Using statin | Not using statin | p Value[a] |
|---|---|---|---|
| | n = 122 | n = 1398 | |
| 14 days post-enrollment | 27/121 (22.3) | 287/1372 (20.9) | 0.807 |
| 2 months post-enrollment | 28/119 (23.5) | 195/1333 (14.6) | 0.014 |
| 6 months post-enrollment | 13/113 (11.5) | 118/1168 (10.1) | 0.759 |
| 12 months post-enrollment | 4/106 (3.8) | 72/1077 (6.7) | 0.196 |
| Last evaluable visit | 5/122 (4.1) | 108/1394 (7.7) | 0.196 |

[a]p value for comparisons between groups was estimated using the normal approximation with continuity correction. $p < 0.05$ was considered significant.

**Table 3. Association between clinical characteristics at enrollment and incomplete recovery.**

| | OR (95% CI)[a] | p Value[b] |
|---|---|---|
| **Statin use (yes vs. no)** | 1.23 (0.68–2.20) | 0.494 |
| **Time** | | |
| **2 months vs. 14 days** | 0.54 (0.43–0.70) | <0.001 |
| **6 vs. 2 months** | 0.48 (0.35–0.64) | <0.001 |
| **12 vs. 6 months** | 0.49 (0.34–0.71) | <0.001 |
| **Sex (male vs female)** | 0.53 (0.38–0.74) | <0.001 |
| **Age** | 1.01 (1.00–1.02) | 0.125 |
| **Solitary EM vs. multiple EM** | 0.55 (0.35–0.87) | 0.009 |
| **Presence of LB-associated constitutional symptoms at enrollment (yes vs no)** | 7.69 (5.54–10.67) | <0.001 |

Abbreviations: OR, odds ratio for incomplete recovery; CI, confidence interval; MEM, multiple erythema migrans; LB, Lyme borreliosis.

[a]Estimated from a multiple logistic regression model with incomplete recovery as the dependent variable, adjusted for a subject variable and study variable as random effects. Each OR is adjusted for all other variables in the table.

[b]$p < 0.05$ was considered significant.

antibiotic treatment was comparable in patients taking statins and in those not taking statins (median 7 days, IQR 4–15 vs. median 7 days, IQR 4–15; $p = 0.9$). The majority ($\geq 76.5\%$) of patients showed complete recovery from 2 months onward, returning to their pre-LB health status. The proportion of patients with incomplete recovery, represented predominantly by the presence of post-LB symptoms, steadily decreased during follow-up regardless of statin use, except at the 2-month visit, when differences between the two groups reached significance (Table 2). At 12 months, 76/1183 (6.4%) patients showed incomplete recovery. When the outcome was evaluated at the last evaluable visit, the proportion of patients with incomplete recovery was similarly low (113/1516, 7.5%). In the multiple logistic regression model, the odds for incomplete recovery were higher for women, for patients with multiple EM, and in those reporting LB-associated constitutional symptoms at enrollment. (Table 3).

## Discussion

In our study of European patients with EM, predominantly infected with *B. afzelii*, statin use was not associated with local progression, dissemination of infection or disease severity as assessed by the size of skin EM, frequency of multiple EM, or presence of LB-associated symptoms, nor with serologic response and borreliae culture positivity, or long-term outcome as assessed by the rate of incomplete recovery. These findings do not support the hypothesis that statins are beneficial in this clinical setting.

Some observational studies suggest that statins may be beneficial in various infections, while the results of a meta-analysis of randomized placebo-controlled trials showed no obvious beneficial effect of statins on risk of infection or infection-related mortality [25]. We are not aware of any clinical study of the effect of statin use on the progression and outcome of early LB, manifesting as EM. However, it was recently reported that statin use is not associated with clinical manifestations, laboratory test results, and outcome in patients with Bannwarth's syndrome [13], which is consistent with our results. Experimental studies, on the other hand, have shown that simvastatin and lovastatin have an inhibitory effect on borrelial growth *in vitro* [8] as well as promote clearance of borreliae in a murine model of LB [12].

Statins may affect pathogenesis of LB through complex lipid interactions and exchanges between spirochete and host cells involving antigenic borrelial outer membrane cholesterol-

glycolipids and other components. These processes may alter borrelial immunogenicity and therefore host immune responses [11].

There is a generally pro-inflammatory cytokine profile in acute LB in murine models and in humans, thought to be important for disease clearance [12, 26]. Persistence of symptoms after treatment of EM may be associated with less intense pre-treatment pro-inflammatory cytokine responses in the skin [27]. However, down-regulation of an initially robust immune response at the proper time appears to be important for optimal resolution of infection [28]. In a murine model, lovastatin, but not simvastatin, was associated with decreased total IgM and IgG and modified cytokine responses to borrelial infection reflecting both $T_H1$ and $T_H2$ immune postures [12]. These responses may have contributed to lovastatin-associated borrelial clearance in that model.

The differential pathophysiology that might explain the divergence of results using statins in *in vitro*, murine model and observational human studies, such as ours, is certainly highly complex. Up regulation of borrelial enzymes, including HMGR-II, under certain conditions (e.g., changes in temperature, pH and increases in extracellular acetate) and some decrease of *in vitro* anti-borrelial effect of simvastatin and lovastatin in experimental studies with HMGR overexpression [8], may play a role among other factors. The binding affinity of statins on HMGR-I present in humans and mice is up to almost $10^4$ fold higher than for HMGR-II found in prokaryotes, including *B. burgdorferi* [29], suggesting a considerable difference in statin inhibitory effect between host and pathogen [10, 30]. Given this difference, it is unclear that the beneficial effects of statins seen in experimental animal models are primarily mediated by their inhibitory effect on borrelial HMGR-II enzymes. Since *B. burgdorferi* is unable to synthesize cholesterol and require cholesterol from their environment [11], it is possible that a statin-associated decrease in host serum cholesterol concentration through HMGR-I inhibition may play a role in attenuating LB infection in murine models. Basal cholesterol levels are lower in mice than in humans [31, 32], among many other metabolic and immunologic differences. In one experimental model, no information was provided on the concentrations of statins and cholesterol in the blood of treated mice [12], nor did we have this information in our study subjects. It is interesting that in another murine model, significant hyperlipidemia associated with certain genetic mutations, such as those affecting cholesterol transport, can lead to more severe presentations of LB [32]. It is not known if a specific level of host cholesterol might be associated with inhibition of borrelial growth; however, the statin doses required to decrease host cholesterol levels such that borrelial growth would be inhibited may be toxic to humans. *In vitro* and murine studies suggest that supra-therapeutic and potentially toxic levels of statins are required to produce antibacterial effects in certain bacterial strains, limiting the clinical value of these findings [12, 30, 33–35].

Some of the statins used in our study may have been more or less effective in inhibiting borrelial growth than simvastatin and lovastatin used in animal studies of infection with *B. burgdorferi*. Only 26.6% of patients in our study were taking one of these two statins. Different statins have been shown to have variable anti-bacterial effects depending on the pathogen studied [34].

In our study, patients taking statins were older and more frequently reported other comorbidities, besides hyperlipidemia. This is consistent with the known increase in the burden of comorbidities with age [36]. Although one study did not find an association between higher age and outcomes in patients with EM [37], it was performed in an American population infected with *B. burgdorferi* and the stratification by age differed from ours, limiting its applicability to our study. A previous European study showed that the presence of LB-associated symptoms at enrollment was the strongest predictor of incomplete recovery in patients with EM, but also that the probability of incomplete recovery increased in older age groups [24].

That study did not find a significant association between comorbidities and clinical outcome suggesting a possible decoupling of age and comorbidities with respect to LB outcome. Furthermore, the use of statins was not detailed, nor was the concentration of lipids measured. In the present study, older patients did not have statistically significant higher odds for incomplete recovery. Although we adjusted the analysis accordingly in the multiple logistic regression model, the observational nature of our study may have introduced a bias that could have masked a positive statin effect in older patients who might otherwise have fared worse. A randomized controlled trial could avoid this pitfall and explore this possibility, but it is unclear how such a trial could be constructed given the semi-random nature of borrelial infection and the probable need for a long duration of prior statin treatment to obtain benefits [38, 39]. Indeed, a study that examined the benefits of *de novo* treatment of infection with statins did not show favorable results [39].

Several other limitations should be considered when interpreting our findings. First, we did not measure serum concentrations of statins, lipids or inflammatory mediators in patients. This might have allowed for a more specific assessment of the association between statin use and borrelial infection in humans. Second, we did not adjust the analysis for the type and dose of statin used. The number of patients receiving a specific dose of a given statin in our study was too small to allow statistical analysis of the association between the type and dose of statin used and selected *B. afzelii* infection characteristics. This is potentially significant, as the greatest benefit may be obtained by matching specific statins and doses with specific infecting agents [34, 40, 41]. Third, patients with multiple EM were over-represented in the analysis, but we recognized this potential bias and adjusted the analysis accordingly. Fourth, our results may relate only to geographic areas where LB is predominantly caused by *B. afzelii*. Fifth, we did not investigate the potential role of statin therapy in various other clinical manifestations of LB. The immunologic, inflammatory, and metabolic mechanisms expressed in other stages and manifestations of LB may be differentially sensitive to statin therapy.

Despite these limitations, we believe that a particular strength of our study was the use of prospectively collected data by infectious disease specialists at a single center with decades of clinical experience and systematic follow-up. Data analysis using a large number of patients with well-characterized manifestations of LB allowed a meaningful comparison of clinical and microbiologic characteristics according to statin use.

In conclusion, statin use was not significantly associated with disease severity, *Borrelia* culture positivity, or long-term outcome in patients with early LB, manifesting as EM and predominantly caused by *B. afzelii*. Future adequately powered and more controlled studies would be needed to clarify whether different doses and types of statins have therapeutic benefit in patients infected with specific *Borrelia* species who present with various manifestations of LB.

## Author Contributions

**Conceptualization:** Daša Stupica.

**Data curation:** Daša Stupica, Fajko F. Bajrović, Stefan Collinet-Adler, Maša Velušček.

**Formal analysis:** Rok Blagus.

**Funding acquisition:** Daša Stupica.

**Investigation:** Daša Stupica, Tjaša Cerar Kišek, Eva Ružić-Sabljić, Maša Velušček.

**Methodology:** Daša Stupica.

**Writing – original draft:** Daša Stupica, Fajko F. Bajrović, Stefan Collinet-Adler.

**Writing – review & editing:** Daša Stupica, Fajko F. Bajrović, Stefan Collinet-Adler.

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
