## [Decision Letter · Decision Letter 0]

22 Oct 2021

PONE-D-21-28253Association between Statin Use and Clinical Course, Microbiologic Characteristics, and Long-Term Outcome of Early Lyme BorreliosisPLOS ONE

Dear Dr. Stupica,

Thank you for submitting your manuscript to PLOS ONE. After careful consideration, we feel that it has merit but does not fully meet PLOS ONE’s publication criteria as it currently stands. Therefore, we invite you to submit a revised version of the manuscript that addresses the points raised during the review process.

The revision must address the statistical issues raised by Reviewer 1 as well as point 5 by Reviewer 2.  Of course, all other reviewer comments must either be addressed in the revision or rebutted in the letter accompanying your revised manuscript.

We look forward to receiving your revised manuscript.

Kind regards,

Sam R. Telford III

Academic Editor

PLOS ONE

Journal Requirements:

 [This research was supported by the Slovenian Research Agency [grant number P3-0296, J3-6788].]

[The authors have no conflicts of interest to declare that are relevant to the content of this article. The funders had no role in the design of the study; in the collection, analyses, or interpretation of data; in the writing of the manuscript, or in the decision to publish the results. Some of the information was presented at the International Symposium on Tick-Borne Pathogens and Disease ITPD 2019, September 2019, Vienna, Austria.]

Additional Editor Comments:

It might be useful to indicate in the conclusion (lines 286 et seq) that your findings are consistent with a general  lack of evidence for an effect of statins on the outcomes of diverse infections (perhaps cite Van den Hoek et al. BMJ 2011)

Reviewers' comments:

Reviewer's Responses to Questions

**Comments to the Author**

1. Is the manuscript technically sound, and do the data support the conclusions?

Reviewer #1: Partly

Reviewer #2: Yes

2. Has the statistical analysis been performed appropriately and rigorously? 

Reviewer #1: No

Reviewer #2: Yes

3. Have the authors made all data underlying the findings in their manuscript fully available?

Reviewer #1: Yes

Reviewer #2: Yes

4. Is the manuscript presented in an intelligible fashion and written in standard English?

Reviewer #1: Yes

Reviewer #2: Yes

5. Review Comments to the Author

Reviewer #1: In this manuscript, Stupica et al examined the association between statin use and clinical course for early Lyme Borreliosis (LB). The study included 1520 prospectively enrolled patients with early LB at a single-center university hospital. The authors did not find any significant association between the statin use and long-term outcome in early LB.

• In line 181, the authors indicated that the difference between the two groups was not significant in the multivariate analysis. However, no multivariate analysis was described in the Methods section, and it is unclear what was included.

• It is unclear why the logistic regression model was used instead of Cox proportional hazards model.

• Eight percent of studied patients received statins treatment at enrollment. Is it because they have underlying health conditions? The authors might also comment the statistical power for the analysis.

Reviewer #2: Review: Association between Statin Use and Clinical Course, Microbiologic Characteristics, and Long-Term Outcome or Early Lyme Borreliosis

This aim of this manuscript was to assess the impact of statins on the clinical course and outcome of patients with early Lyme borreliosis. The strengths of this study are a large patient cohort (n>1300) evaluated at a single medical center, and the systematic follow-up evaluation of these patients for 1 year. The manuscript is well written, and the conclusions are clear. I have only minor comments which are provided below.

1) In the Methods section (line 06-99), the authors make a distinction in defining “partial recovery” vs “treatment failure”, however no distinction is made between these groups in the manuscript. The rationale for this distinction is not clear. The groups could be combined into “incomplete recovery” in the Methods section, or additional explanation is needed for why the distinction is necessary. Did the authors find a difference in patients with incomplete recovery were stratified by “partial” vs “failure”?

2) Line 112-113. It would help to state that the second skin biopsy was taken 2-3 months after the start of antibiotic therapy.

3) Please clarify that Table 3 is the association of clinical characteristics at first visit, before antibiotics. This could go into the title for Table 3: Association between clinical characteristics at enrollment and incomplete recovery.

4) Line 195-197: This section is a better fit for the paragraph discussing the results in Table 2. It could follow the sentence on line 181.

5) Line 214-217: Although reference 12, which is based on mice, supports the upregulation of inflammatory responses to statins, references 25 and 26, which are based in humans, do not. Indeed, cytokine levels in patients with EM are generally higher at enrolment during acute infection than after antibiotic therapy / convalescence which goes along with the role of cytokines in the pathogenesis of this disease. One could argue that the greater inflammatory responses to statins could be advantageous and lead to Borrelia clearance, but they could also be disadvantageous by contributing to immune-associated symptoms. It is not clear how these points connect back with the idea that lipids may impact Borrelia immunogenicity. The paragraph should be rewritten and/or better references are needed in support of this point.

6. PLOS authors have the option to publish the peer review history of their article (what does this mean?). If published, this will include your full peer review and any attached files.

Reviewer #1: No

Reviewer #2: No

---

## [Author Response · Author response to Decision Letter 0]

31 Oct 2021

Dear Prof. Telford

Please find enclosed a rewritten version of our manuscript entitled “Association between Statin Use and Clinical Course, Microbiologic Characteristics, and Long-Term Outcome of Early Lyme Borreliosis. A Post Hoc Analysis of Prospective Clinical Trials of Adult Patients with Erythema Migrans”. We are grateful to the reviewers for their thoughtful comments and suggestions, enabling us to reconsider the issues they have raised and rewrite the manuscript in line with their comments. Our detailed responses are listed below.

Authors' reply to the Review Report

Journal Requirements:

Answer: We have modified the manuscript to meet PLOS ONE’s requirements.

2. Thank you for stating the following financial disclosure: [This research was supported by the Slovenian Research Agency [grant number P3-0296, J3-6788]. Please state what role the funders took in the study. If the funders had no role, please state: "The funders had no role in study design, data collection and analysis, decision to publish, or preparation of the manuscript." 

Answer: Corrected. Please see lines 392-393.

[The authors have no conflicts of interest to declare that are relevant to the content of this article. The funders had no role in the design of the study; in the collection, analyses, or interpretation of data; in the writing of the manuscript, or in the decision to publish the results. Some of the information was presented at the International Symposium on Tick-Borne Pathogens and Disease ITPD 2019, September 2019, Vienna, Austria.]

Answer: Corrected. Please see the cover letter.

The minimal dataset required to replicate the reported study findings in their entirety are within the paper and its Supporting information files. Additional data cannot be shared publicly because of potentially identifying/sensitive patient information.

Answer: Corrected. Please see the cover letter.

Answer: Corrected. Please see lines 147-161.

Answer: We are not aware that we cited any retracted references. We have added two new references (Ref No 25 and Ref No 28) that have been highlighted in the revised manuscript.

Additional Editor Comments:

It might be useful to indicate in the conclusion (lines 286 et seq) that your findings are consistent with a general lack of evidence for an effect of statins on the outcomes of diverse infections (perhaps cite Van den Hoek et al. BMJ 2011)

Answer: We have positioned our results within the body of studies exploring statin effects on infections in lines 269-271. We felt that it fit the flow of the article better at this point in the discussion rather than in the conclusion.

Reviewer #1: In this manuscript, Stupica et al examined the association between statin use and clinical course for early Lyme Borreliosis (LB). The study included 1520 prospectively enrolled patients with early LB at a single-center university hospital. The authors did not find any significant association between the statin use and long-term outcome in early LB.

• In line 181, the authors indicated that the difference between the two groups was not significant in the multivariate analysis. However, no multivariate analysis was described in the Methods section, and it is unclear what was included.

Answer: The multivariate analysis we are referring to the multiple logistic regression model, which is described in the Methods section. The multivariate model was performed to control for selected confounders listed in the Statistical methods section and in the Discussion section. The paper was revised so that multiple logistic regression is used instead of multivariate analysis throughout the paper. Please see lines 236 and 351-352.

• It is unclear why the logistic regression model was used instead of Cox proportional hazards model.

Answer: We observed patients at fixed, pre-defined time points (visits), so there is no way to know exactly when the event (incomplete recovery) occurred between visits. Therefore, we can only model the probability of an event occurring within a rather large window of time, hence the use of logistic regression. We accounted for the fact that the same individuals were measured repeatedly in time by including a random effect.

• Eight percent of studied patients received statins treatment at enrollment. Is it because they have underlying health conditions? The authors might also comment the statistical power for the analysis.

Answer: Patients on statins at enrollment had hyperlipidemia and/or comorbidities requiring primary or secondary prophylaxis of cardiovascular disease. We added these details in lines 93-94. A formal power calculation was not felt to be appropriate for this post-hoc analysis. However, the variance of the Bernoulli random variable decreases as the event proportion is smaller (or larger) than 0.5 which in our case increases the power of our study. At the same time the number of events (122) is large enough so that we meet the recommended number of events per variable ratio (EPV>10) in our multiple logistic regression model. 

Reviewer #2: Review: Association between Statin Use and Clinical Course, Microbiologic Characteristics, and Long-Term Outcome or Early Lyme Borreliosis

This aim of this manuscript was to assess the impact of statins on the clinical course and outcome of patients with early Lyme borreliosis. The strengths of this study are a large patient cohort (n>1300) evaluated at a single medical center, and the systematic follow-up evaluation of these patients for 1 year. The manuscript is well written, and the conclusions are clear. I have only minor comments which are provided below.

1) In the Methods section (line 06-99), the authors make a distinction in defining “partial recovery” vs “treatment failure”, however no distinction is made between these groups in the manuscript. The rationale for this distinction is not clear. The groups could be combined into “incomplete recovery” in the Methods section, or additional explanation is needed for why the distinction is necessary. Did the authors find a difference in patients with incomplete recovery were stratified by “partial” vs “failure”?

Answer: We agree with the reviewer that this distinction, a holdover from related studies, can be confusing here and feel that it is not necessary for this manuscript. We have therefore combined these two outcomes into one. Please see lines 104-105 and 108.

2) Line 112-113. It would help to state that the second skin biopsy was taken 2-3 months after the start of antibiotic therapy.

Answer: Corrected. Please see line 120.

3) Please clarify that Table 3 is the association of clinical characteristics at first visit, before antibiotics. This could go into the title for Table 3: Association between clinical characteristics at enrollment and incomplete recovery.

Answer: Corrected. Please see line 249.

4) Line 195-197: This section is a better fit for the paragraph discussing the results in Table 2. It could follow the sentence on line 181.

Answer: Corrected. Please see lines 234-236.

5) Line 214-217: Although reference 12, which is based on mice, supports the upregulation of inflammatory responses to statins, references 25 and 26, which are based in humans, do not. Indeed, cytokine levels in patients with EM are generally higher at enrolment during acute infection than after antibiotic therapy / convalescence which goes along with the role of cytokines in the pathogenesis of this disease. One could argue that the greater inflammatory responses to statins could be advantageous and lead to Borrelia clearance, but they could also be disadvantageous by contributing to immune-associated symptoms. It is not clear how these points connect back with the idea that lipids may impact Borrelia immunogenicity. The paragraph should be rewritten and/or better references are needed in support of this point.

Answer: We agree with the reviewer that this paragraph, as initially written, was unclear. We have divided the paragraph into two paragraphs to keep the two subjects (immunogenicity and the interplay of TH1 and TH2 cytokines) apart. In the second paragraph, we have hopefully more clearly highlighted some modifications of immune mechanisms by which statins may possibly impact borrelial clearance in the murine model cited. Please see lines 292-300.

---

## [Decision Letter · Decision Letter 1]

25 Nov 2021

Association between Statin Use and Clinical Course, Microbiologic Characteristics, and Long-Term Outcome of Early Lyme Borreliosis

PONE-D-21-28253R1

Dear Dr. Stupica,

We’re pleased to inform you that your manuscript has been judged scientifically suitable for publication and will be formally accepted for publication once it meets all outstanding technical requirements.

Kind regards,

Sam R. Telford III

Academic Editor

PLOS ONE

Additional Editor Comments (optional):

Reviewers' comments:

Reviewer's Responses to Questions

**Comments to the Author**

1. If the authors have adequately addressed your comments raised in a previous round of review and you feel that this manuscript is now acceptable for publication, you may indicate that here to bypass the “Comments to the Author” section, enter your conflict of interest statement in the “Confidential to Editor” section, and submit your "Accept" recommendation.

Reviewer #1: All comments have been addressed

2. Is the manuscript technically sound, and do the data support the conclusions?

Reviewer #1: Yes

3. Has the statistical analysis been performed appropriately and rigorously? 

Reviewer #1: Yes

4. Have the authors made all data underlying the findings in their manuscript fully available?

Reviewer #1: Yes

5. Is the manuscript presented in an intelligible fashion and written in standard English?

Reviewer #1: Yes

6. Review Comments to the Author

Reviewer #1: (No Response)

7. PLOS authors have the option to publish the peer review history of their article (what does this mean?). If published, this will include your full peer review and any attached files.

Reviewer #1: No

---

## [Editor Report · Acceptance letter]

6 Dec 2021

PONE-D-21-28253R1 

Association between Statin Use and Clinical Course, Microbiologic Characteristics, and Long-Term Outcome of Early Lyme Borreliosis. A Post Hoc Analysis of Prospective Clinical Trials of Adult Patients with Erythema Migrans 

Dear Dr. Stupica:

I'm pleased to inform you that your manuscript has been deemed suitable for publication in PLOS ONE. Congratulations! Your manuscript is now with our production department. 

Kind regards, 

on behalf of

Dr. Sam R. Telford III 

Academic Editor

PLOS ONE